# Beta-Hydroxybutyrate (BHB), Glucose, Insulin, Octanoate (C8), and Decanoate (C10) Responses to a Medium-Chain Triglyceride (MCT) Oil with and without Glucose: A Single-Center Study in Healthy Adults

**DOI:** 10.3390/nu15051148

**Published:** 2023-02-24

**Authors:** Christina Heidt, Manfred Fobker, Mary Newport, Reinhold Feldmann, Tobias Fischer, Thorsten Marquardt

**Affiliations:** 1Department of Pediatrics, University Hospital Muenster, 48149 Muenster, Germany; 2Centre of Laboratory Medicine, University Hospital Muenster, 48149 Muenster, Germany; 3Spring Hill Neonatology, Inc., Spring Hill, FL 34610, USA; 4Department of Food, Nutrition and Facilities, FH Muenster, University of Applied Sciences, 48149 Muenster, Germany

**Keywords:** medium-chain fatty acids, beta-hydroxybutyrate, glucose, insulin, octanoic acid, decanoic acid, ketogenic diet, cognitive function

## Abstract

MCTs are increasingly being used to promote ketogenesis by patients on ketogenic diet therapy, but also by people with other conditions and by the general public for the perceived potential benefits. However, consumption of carbohydrates with MCTs and untoward gastrointestinal side effects, especially at higher doses, could decrease the sustainability of the ketogenic response. This single-center study investigated the impact of consuming carbohydrate as glucose with MCT oil compared to MCT alone on the BHB response. The effects of MCT oil versus MCT oil plus glucose on blood glucose, insulin response, levels of C8, C10, BHB, and cognitive function were determined, and side effects were monitored. A significant plasma BHB increase with a peak at 60 min was observed in 19 healthy participants (24.4 ± 3.9 years) after consuming MCT oil alone, and a more delayed but slightly higher peak was observed after consuming MCT oil plus glucose. A significant increase in blood glucose and insulin levels occurred only after MCT oil plus glucose intake. The overall mean plasma levels of C8 and C10 were higher with the intake of MCT oil alone. MCT oil plus glucose consumption showed improved scores for the arithmetic and vocabulary subtests.

## 1. Introduction

Medium-chain triglycerides (MCTs) are fats primarily made up of fatty acids with carbon chain lengths ranging from C6-C12 [1]. MCTs are present in natural substances like coconut and in small amounts in dairy products such as butter and goat milk [2,3,4,5,6]. Human breast milk also contains medium-chain fatty acids (MCFAs) [7]. MCTs in commercial products are often a mixture of caprylic acid (C8) and capric acid (C10). Exogenous MCTs are known to promote ketogenesis because of their fast absorption via the portal vein and their oxidation to acetyl-CoA in the liver [8]. In contrast to long-chain dietary fatty acids (LCFAs), which enter the cell via specific transport proteins like CD36 and FATPs (fatty acid transport proteins) and have to build carnitine esters for import into the mitochondrial matrix, medium-chain fatty acids (MCFAs) do not rely on membrane transporters for uptake into cells and can be directly transported into the mitochondrial matrix without the carnitine shuttle [9]. All fatty acids undergo β-oxidation in order to produce acetyl-CoA, which will be further metabolized in the citric acid cycle. When the citric acid cycle is overloaded, excess acetyl-CoA is used for the synthesis of ketone bodies, mainly β-hydroxybutyrate (BHB) and, to a lesser extent, acetoacetate (AcAc), which can be transported to the brain, muscle, or to other organs, where they can be converted back to acetyl-CoA and used in the citric acid cycle to produce ATP, as shown in Figure 1 [10,11]. Basically, ketones serve as a storage form of acetyl-CoA when the citric acid cycle is overloaded. Aside from serving as fuel to non-hepatic tissues, studies have indicated that MCT supplementation of a regular diet appears to affect cognitive function by improving memory, language, executive function, and processing speed [12,13,14,15,16]. These improvements strongly correlate with the increase in the production of BHB generated after a single dose of MCT or with regular consumption over several months [17,18]. The combination of MCT and carbohydrate intake could potentially inhibit ketosis through an increase in insulin secretion. A few studies have demonstrated that sucrose or an accompanying meal containing carbohydrates can influence the ketogenic response of MCTs. Consuming carbohydrates after MCTs also appears to decrease their ketogenic effect [17,19,20]. Gastrointestinal discomfort, a common problem after MCT intake, could decrease the tolerability and sustainability of their ketogenic potential as well [21]. Taking MCT with meals is known to improve gastrointestinal tolerability [22,23]. This study sought to investigate the BHB response to MCT oil by evaluating the impact of adding glucose (0.2 g/kg of body weight) to the MCT intake (0.5 g/kg of body weight) and measuring blood glucose, insulin response, total ketones, BHB, AcAc, and plasma levels of C8 and C10. The second aim of this study was to assess tolerance and satiety by intermittently monitoring for side effects and satiety at different time points for MCT alone versus MCT plus glucose for each participant. Additionally, we evaluated the impact of MCT oil with and without glucose on cognitive function. The findings in this study could inform and improve study design, such as meal macronutrient composition, and the interpretation of results in future clinical trials that aim to achieve sustained ketosis by adding MCTs (e.g., a classical ketogenic diet therapy in drug-resistant epilepsy).

LCFAs: long-chain fatty acids; MCFAs: medium-chain fatty acids; CD36: [also known as fatty acid translocase (FAT), glycoprotein IIIb (GPIIIb), or glycoprotein IV] is a member of the class B2 scavenger receptor family; FATPs: fatty acid transport proteins; CPT1: carnitine palmitoyltransferase 1; CPT2: carnitine palmitoyltransferase 2; BHB: β-hydroxybutyrate; AcAc: acetoacetate; CAC: citric acid cycle; HMG-CoA: 3-hydroxy-3-methylglutaryl-CoA.

## 2. Materials and Methods

### 2.1. Participants

Nineteen healthy volunteers (12 females and 7 males) were recruited by advertising at the university hospital in Muenster (UKM), Germany, and gave written, informed consent to participate in this crossover study. Inclusion criteria were age 18–30 years, being non-overweight and non-obese (≤25 BMI), not being diagnosed as diabetic, not currently or previously following a ketogenic diet, and not consuming any MCT-based supplements (MCT oil or coconut oil). Baseline characteristics of the participants are described in Table 1. Ethical approval for this study was obtained from the local medical ethics committee in Muenster (approval number: 2018-616-f-S), and the study was designed and conducted in accordance with the Declaration of Helsinki (2013). The study was registered as a clinical trial on DRKS.de (DRKS-ID: DRKS00026605).

### 2.2. Test Ingredients (MCT and Glucose)

For test day 2 (MCT intervention), MCT was provided to participants using a commercially available MCT oil (Dr. Schär AG, Italy) with a 60:40 ratio of C8:C10 (0.5 g MCT/kg of body weight in 200 mL of still drinking water). We used this dosage as it was most likely to elicit an effect without unduly exposing participants to gastrointestinal discomfort, which was more likely to arise with higher MCT intake [13]. For test day 3 (MCT plus glucose), glucose (Glucose-Monohydrate, EDQM) was added to the MCT oil at a dosage of 0.2 g glucose/kg of body weight in 200 mL of still-drinking water. We used this dosage based on ketogenic diet protocols, which were often set in the range of 10–60 g carbohydrates per day [24]. Control drink (test day 1) was 200 mL of still-drinking water, as summarized in Figure 2.

### 2.3. Experimental Procedure

This was a single-center study carried out at UKM, Germany. The study followed a repeated-measures design involving three separate test days at the exact same time during four consecutive weeks. All three study days were identical for each participant, as shown in Figure 2. On each study day, participants arrived at 08:00 a.m. after a 12 h overnight fast and a minimum of 24 h without alcohol intake. Details of pre-study meals and drinks were recorded by each participant using a 3-day questionnaire and monitored and calculated, using PRODI software, at the study’s completion by a dietician. Participants were encouraged to eat a similar pre-study dinner on each study day. A forearm venous catheter was installed, a baseline blood sample was withdrawn, and all participants began consuming the test drink, which was provided in a glass, within the next 2 to 5 min at a time point defined as t0. The consumption of the study drink was directly supervised by the physician and was consumed in a single drink of 200 mL of still drinking water in five to eight swallows (less than 60 s). Test drink one (control), test drink two (MCT), and test drink three (MCT plus glucose) were mixed with a spoon immediately prior to consumption. Participants were blinded to the test drinks. After finishing the study drink, the participants were only allowed to consume drinking water. Blood samples were taken every 30 min during the first 2 h (t0–t3) and every 60 min during the next 3 to 5 h (t4–t7).

### 2.4. Plasma Metabolite Analyses

Blood samples were centrifuged for 10 min at 4 °C and plasma was stored at −80 °C until analyzed. Glucose, insulin, ketones, and C8 and C10 were measured in accordance with routine procedures at the university hospital lab. Plasma glucose was analyzed by using the enzymatic hexokinase method with an automated clinical chemistry analyzer, Cobas c702 (Roche Diagnostics GmbH, Mannheim, Germany). Plasma insulin was measured by an electrochemiluminescence immunoassay on a Cobas e801 analyzer (Roche Diagnostics GmbH, Mannheim, Germany). Plasma β-HB, AcAc, total ketones were quantified enzymatically using the Total Ketone and BHB autokit (FUJIFILM Wako Chemicals Europe GmbH, Neuss, Germany) on a Cobas integra 400 plus analyzer (Roche Diagnostics GmbH, Mannheim, Germany). The determination of plasma C8 and C10 fatty acid levels by gas chromatography-mass spectrometry was performed according to a modified method of Bartolucci et al. [25]. For this purpose, 100 µL serum was mixed with 10 µL internal standard (20 µg/mL D3-Octanoic acid in methanol), 20 µL 37% HCl/H_2_O 1:1 (*v/v*), 200 µL H20, 100 µL Methyl-t-butyl ether, and 50 µL n-hexan in a 0.5 mL Eppendorf reaction vessel and centrifuged (3 min, 10,000 rpm). The supernatant (50 µL) was transferred to an autosampler vial with a conical insert, and 1 µL was loaded in splitless mode onto a GC-MS (QP2010 Ultra; Shimadzu, Duisburg, Germany) equipped with a PTV injector and a capillary column (Stabilwax-DA, 15 m length, 0.25 mm internal diameter, and 0.25 µm film thickness; Restek GmbH, Bad Homburg, Germany) and an autosampler AS AOC-20 s. Measuring time: 58.7 min; injector temperature: 70 °C; column oven temperature program isothermal: 2 min at 40 °C; gradient 12°C/min to 240 °C; isothermal 40 min at 240 °C. The carrier gas was helium with a constant flow of 1 mL/min. The transfer line to the ion source was held at 260 °C. The MS detector was set to selected ion monitoring (SIM) mode as *m*/*z* 60 (C8/C10) and *m*/*z* 63 (D3-Octanoic acid, internal standard). An example base peak intensity chromatogram of free fatty acids is presented in Appendix A. LabSolutions software (Version 2.72; Shimadzu, Duisburg, Germany) was used for GC–MS data acquisition and processing.

### 2.5. Questionnaire on Satiety and Tolerability 

A questionnaire on satiety and tolerance was used immediately after t1 and t7 on each study day. The questionnaire was designed for this study, using a 0–10-point scale, from 0 = not satiated at all to 10 = very satiated. At the end of the questionnaire, participants were asked to record any side effects (e.g., bloating, nausea, diarrhea), and any symptoms were recorded hourly during testing.

### 2.6. Cognitive Test

Cognitive performance was measured using the Wechsler Adult Intelligence Scale, 4th edition (WAIS-IV). The WAIS-IV is a classic tool to assess general intelligence as well as four subcategories, including verbal comprehension, fluid reasoning, working memory, and processing speed. In the current study, full WAIS-IV assessment was not employed due to the long completion time. We used two “vocabulary” subtests as part of the Verbal Comprehension Index (VCI), which is designed to measure verbal reasoning and concept formation. The vocabulary subtest required the participant to try to define up to 30 words. This test was performed at t2. The second subtest was “arithmetic,” part of the Working Memory Index (WMI), which was designed to assess the ability to sustain attention, concentrate, and exert mental control. This subtest consisted of 22 timed arithmetic questions to be solved without the use of pencil and paper, such as: “If Jo has 12 buns, he then eats 3 and gives 4 away; how many does he have left?” The arithmetic subtest was performed at t7. The vocabulary and arithmetic subtests were chosen because they are close to everyday cognitive performance and easy to administer, both loading highly on their underlying factors “verbal comprehension” (vocabulary) and “working memory” (arithmetic), with arithmetic being largely independent from verbal comprehension according to Ward et al. [26]. 

Both subtests were performed under the supervision of a physician (time needed to administer the subtest: 10 min per subtest), as shown in Figure 2. [27].

### 2.7. Statistical Analysis

Statistical analyses were performed using GraphPad Prism version 9.3.1 (GraphPad, La Jolla, CA, USA) and the R statistical programming language (R script; see Appendix A). The normality of the analyzed variables was tested with the Kolmogorov–Smirnov test at the level of *p* < 0.05. Metabolic data (plasma metabolites) did not follow a normal distribution and were therefore analyzed using non-parametric methods; the non-parametric Mann–Whitney-U/Wilcoxon test was used. The level of statistical significance was set at *p* < 0.05. Spearman’s correlation test was used for correlations. *p* < 0.05 was considered to indicate a statistically significant difference. Linear regression analysis was carried out to determine the relationship between plasma C8 and C10 levels and plasma BHB levels by using the statistical programming language R version 4.2.1.

## 3. Results

Nineteen healthy volunteers [12 females (63%), 7 males (37%)] completed the study in full. Total median MCT intake was 32 (30–35.5) g during the MCT oil intervention, and total median glucose intake was 12.7 (11.8–13.8) g during the MCT oil plus glucose intervention. Nutritional analysis of reported pre-study meals and drinks (carbohydrates, protein, fat, and calories) on each test day, calculated by a dietician, confirmed no differences in results for the individual participants (Appendix A). In all cases, the percent of carbohydrate intake in pre-study meals was more than 50%, which is more typical of a habitual non-ketogenic diet.

### 3.1. Beta-Hydroxybutyrate 

#### 3.1.1. Responses to MCT Oil Alone

The median (IQR) BHB at baseline was 50.8 (30.7–105.7) µmol/L, which increased to 141.2 (49.9–241.4) after 30 min, to 348.8 (204.4–408.4) µmol/L at 120 min, to 417.8 (266.4–473.7) µmol/L at 180 min, and to 465.4 (354.8–509.3) µmol/L at 240 min (Figure 3). Significant differences in plasma BHB in MCT oil intervention compared to control were observed at t2 (*p* = 0.003), t3 (*p* = 0.005), t4 (*p* = 0.004), t5 (*p* = 0.006), t6 (*p* = 0.005), and t7 (*p* = 0.03). 

#### 3.1.2. MCT Oil Alone versus MCT Plus Glucose

The difference in BHB levels was not statistically significant between MCT oil alone and MCT oil plus glucose at 30 min (*p* = 0.11); however, the increase in BHB levels from MCT oil alone was significantly greater at 60 min compared to MCT oil plus glucose (*p* = 0.0004), and levels of BHB were similar at 120 min for MCT alone and MCT plus glucose (*p* = 0.81). Thus, the ketogenic response of MCT plus glucose was delayed compared to MCT alone but reached similar final concentrations. 

Plasma AcAc levels and total ketone levels are presented in the Appendix A. 

#### 3.1.3. Responses to MCT Oil Plus Glucose 

The median (IQR) BHB at baseline was 73.8 (27.9–192.3) µmol/L, which increased to 374.9 (235.1–511.8) µmol/L at 120 min, 381.7 (252.6–644.2) µmol/L at 180 min, 453.7 (357.2–722) µmol/L at 240 min and 524.6 (377–617) at 300 min (Figure 3). Significant differences in plasma BHB in MCT oil plus glucose intervention compared to control were at t4 (*p* = 0.014), t5 (*p* = 0.0001), t6 (*p* = 0.0001), and t7 (*p* = 0.001). 

### 3.2. Glucose Responses

#### Responses to MCT Oil Alone

The median (IQR) glucose level at baseline was 72 (65–76) mg/dL, which slightly increased with prolonged fasting throughout the experimental period (Figure 4A).

For responses to MCT oil plus glucose, the median (IQR) glucose at baseline was 68 (65–76) mg/dL, which significantly increased to 93 (85–111) mg/dL after 30 min (*p* = 0.0005). The median glucose level then decreased to 64 (55–72) mg/dL at 60 min and increased to 68 (63–71) mg/dL at 90 min (Figure 4A). 

### 3.3. Insulin Responses

#### 3.3.1. Responses to MCT Oil Alone

The median (IQR) insulin level was 4.3 (2.3–6 µU/mL) at baseline, which increased to 5.1 (1.9–6.7) µU/mL after 30 min; however, this difference was not statistically significant compared to the control intervention (*p* = 0.8). Median insulin fell to 4.7 (2.9–5.9) µU/mL by 60 min. The lowest median insulin was measured at 2.5 (0.5–3.4) µU/mL after 300 min (Figure 4B).

#### 3.3.2. Responses to MCT Oil Plus Glucose 

The median (IQR) insulin level was 4.7 (3.4–7) µU/mL at baseline, which significantly increased to 17.9 (12.6–28.1) µmol/L after 30 min compared to MCT oil intake (*p* = 0.0003). Median insulin decreased to 4.1 (2.7–7) µU/mL at 120 min (Figure 4B). Insulin levels were significantly higher compared to MCT alone at 240 (*p* = 0.002) and 300 min (*p* = 0.04).

### 3.4. Plasma C8 and C10 Levels and Area under the Curve for C8 and C10 Response

Plasma octanoate (C8) and decanoate (C10) levels were not measured during the control intervention, which did not contain C8 or C10. Plasma octanoate (C8) and decanoate (C10) concentrations during the MCT oil and MCT plus glucose interventions are presented in Figure 5.

#### 3.4.1. Responses to MCT Oil Alone

The median (IQR) C8 level was 1.2 (1–1.5) µg/mL at baseline, which increased to 4.5 (1.8–12) µg/mL after 30 min and to 16.52 (7.4–27.8) µg/mL at 60 min. The median (IQR) C10 level was 0.85 (0.68–0.94) µg/mL at baseline, which increased to 1.5 (1.8–12) µg/mL after 30 min, to 2.64 (1.46–4.08) µg/mL at 60 min and to 3.21 (1.79–51.5) µg/mL at 90 min. 

#### 3.4.2. Responses to MCT Oil Plus Glucose 

The median (IQR) C8 level was 1.2 (0.9–1.5) µU/mL at baseline, which increased to 2.1 (1.5–6.5) µg/mL after 30 min and to 9.6 (2.3–13.8) µg/mL at 60 min. The highest median C8 level was measured at 17.1 (9.6–21.5) µg/mL after 240 min. The median (IQR) C10 level was 0.76 (0.67–0.88) µg/mL at baseline, which slightly increased to 0.97 (0.68–1.61) µg/mL after 30 min, to 1.29 (0.8–2.2) µg/mL at 60 min, and to 2.31 (1.3–3.5) µg/mL at 90 min. The highest median C10 level was measured at 3.7 (3.1–6) µg/mL after 300 min.

The area-under-the-curve (AUC) for plasma BHB is shown in Figure 6. There was a significant difference in BHB AUC with MCT and MCT + Glc interventions in comparison to control. There was no significant difference in BHB AUC between MCT and MCT plus glucose intervention.

The area-under-the-curve (AUC) for C8 and C10 is shown in Figure 7. No significant differences between AUC C8 and AUC C10 were observed between MCT oil alone and MCT oil plus glucose intervention during the study day (t0–t7). Correlations between AUC and metabolic data are presented in the Appendix A. 

A linear regression analysis between plasma BHB and plasma C8 and C10 is presented in Figure 8A,B. Linear regression analysis with plasma C8 showed a significant increase in plasma BHB levels with MCT (slope = 7.587, t = 7.040, *p* < 0.001) and a significantly greater increase with MCT plus glucose (slope = 15.594, t = 10.698, *p* < 0.001).

Linear regression analysis with plasma C10 showed a significant increase in plasma BHB levels with MCT (slope = 30.47, t = 4.97, *p* < 0.001) and a significantly greater increase with MCT plus glucose (slope = 42.334, t = 5.882, *p* < 0.001).

### 3.5. Satiety and Side Effects

#### 3.5.1. Satiety

The median satiety score was 5 (0 = not satiated at all, to 10 = very satiated) after MCT oil alone and MCT oil plus glucose ingestion at t1. The median satiety score decreased to 3 (a little bit satiated) in both interventions at t7. In the control intervention, the median satiety score decreased from 3 (a little bit satiated) at t1 to 1 (not satiated) at t7 (data not presented). 

#### 3.5.2. Side Effects

Five participants (26.3%) experienced side effects. None of these symptoms were reported by the participants to be severe enough for study discontinuation. The most common side effects are summarized in Table 2. 50% fewer side effects were reported during the MCT oil plus glucose intervention compared to MCT oil alone. In all cases, all the symptoms were experienced after the MCT oil and MCT oil plus glucose ingestion and resolved completely within 30 (t1) to 60 (t2) minutes. No side effects were reported after taking the control drink. 

### 3.6. Cognitive Assessment 

#### 3.6.1. Subtests “Vocabulary”

For the vocabulary subtest, there was no significant difference between the control and the MCT oil alone intervention, nor between MCT oil alone and MCT oil plus glucose. However, a significant difference was found between the control and the MCT oil plus glucose intervention (*p* = 0.005); see Figure 9A.

#### 3.6.2. Subtest “Arithmetic”

For the arithmetic subtest, there was no significant difference between the control and the MCT oil alone intervention. However, a significant difference was found between the control and the MCT oil plus glucose intervention (*p* = 0.014); see Figure 9B. 

## 4. Discussion

Nutritional ketosis with a ketogenic diet (KD) alone can be difficult to maintain indefinitely, due to the need for constant attention to meal planning, weighing, and measuring, the very low carbohydrate content, and common side effects like constipation, unintended weight loss, and muscle cramps [28]. To address this, researchers have developed a strategy to induce and maintain ketosis by adding MCT oil to a ketogenic diet, which may allow for more carbohydrate content and variety in meals, thereby potentially improving long-term compliance [29,30,31,32,33]. MCT supplementation of 10–15 g can increase blood ketone levels to 0.3 to 1.0 mmol/L [23,34,35]. 

The results of this study confirmed these previous findings. MCT oil ingestion led to elevated BHB levels (>0.5 mmol/L) in many of the participants. The highest individual BHB level was 0.9 mmol/L following a single dose of MCT oil alone. MCT oil supplementation at 0.5 g per kg of body weight in healthy participants significantly increased plasma BHB levels compared to the control intervention. The increase in ketosis was faster after intake of MCT oil alone compared to MCT oil plus glucose during the first 90 min but reached similar levels for the remainder of the study. The highest individual BHB level after MCT oil plus glucose peaked at 1 mmol/L at 300 min.

Careful consideration of administering large doses of MCT together with carbohydrate is warranted, since the insulin response related to carbohydrate intake activates acetyl CoA-carboxylase, which increases the cellular concentration of malonyl-CoA, which could in turn result in more acetyl-CoA being redirected away from ketogenesis and toward lipogenesis, less oxidation of LCFAs, and more LCFAs available for esterification to form TG and cholesterol esters; see Figure 1 [36]. This effect has been attributed to competition between the fatty acid and glucose oxidations, and the magnitude of activation of acetyl-CoA carboxylase increases with chain length from C6 to C12 but decreases again with fatty acids of longer chain length [8].

Only a few studies have investigated the effect of an accompanying meal or carbohydrate content on ketone production from MCT intake [36]. These studies show that when MCTs were mixed with other foods, the ketogenic effect was decreased [37]. Norgren et al. compared the ketogenic effect with and without the addition of 50 g of glucose to 20 g of C8 in 15 healthy older adults (65–73 years) and found a 63% decrease in ketone [37]. Norgren et al. compared the ketogenic effect with and without the addition of 50 g of glucose to 20 g of C8 in healthy older adults and found a 63% decrease in ketone production [38]. 

In another study, St-Pierre et al. showed that, when 20 mL (18.9 g) of MCT were given to nine healthy volunteers (mean age 34 ± 12 years), diluted in lactose-free skim milk, the ketogenic response was 2-fold stronger if the drink was given without an accompanying meal containing 55 g of carbohydrate [39]. 

Freemantle et al. examined the ketogenic effect in 32 healthy adults in three different age groups (young = mean age 23 ± 1; middle-aged = 50 ± 1; and elderly = 76 ± 2 years), providing a very low carbohydrate (3 g) meal with MCT (71 g ± 4 g). The group measured plasma BHB levels hourly and found an increase in BHB level to 0.7 mmol/L at 1 and 2 h after administration, with a peak of 1.3 mmol/L at 6 h [40]. 

The results of this study, that consuming a low-carbohydrate meal with MCT does not inhibit the ketogenic effect but slightly reduces the production of BHB, are in line with the findings of previous studies. It should be noted that this study was limited to healthy young adults aged 24.4 ± 3.9 years, whereas the other studies included adults ranging from young to elderly. The study also provides additional information, specifically that consumption of MCT (0.5 g/kg BW) with a small amount of carbohydrate (0.2 g/kg BW), typically prescribed in ketogenic diet therapy, may prolong the time required to attain the maximal plasma BHB concentration. Quantification of ketosis in a longer study day (>6 h) with higher amounts of monosaccharides, mixtures of different carbohydrates, meals, or snacks could be the focus of future studies to determine how much flexibility is possible in the implementation of a ketogenic diet with the addition of MCT oil. 

These new results differ from a published study by Vandenberghe et al., who assessed the influence of MCT on metabolic switching from glucose to ketones as a fuel in 10 young adults (mean age 28 ± 7 years) and in 10 older adults (mean age 65 ± 6 years) [41]. The authors reported an increased plasma ketone response in 30 min after 10 g MCT intake with a low carbohydrate meal containing 3 g carbs and after 10 g MCT intake with a carbohydrate-rich meal containing 49 g carbs. The increase in plasma ketones after MCT intake was essentially independent of carbohydrate intake [41]. The inconsistency could arise from using a liquid emulsion, as the MCT composition (60% octanoic and 40% decanoic acid) was the same. Courchesne-Loyer et al. showed that the emulsification of an MCT drink increased the BHB response after 30 min compared with a non-emulsified MCT drink [22]. It should also be noted that emulsification of MCT with liquids using a high-pressure homogenizer may also reduce gastrointestinal discomfort, especially diarrhea, a key side effect observed particularly at the 30 g dose of MCT, and increase the ketogenic effect by ≤4-fold [22]. However, the effect of emulsifying MCTs and the optimal technique need further investigation [42]. In the current study, the MCT doses were manually stirred, and two episodes of diarrhea were reported in each MCT intervention affecting the same participants. 

Taking MCT with meals is known to improve gastrointestinal tolerability [11,43]. Similar results were observed in this study. Participants experienced 50% fewer side effects when consuming MCT with glucose compared to MCT alone. Side effects did not appear to be related to dose or BHB concentrations. 

A slight decrease in blood glucose was observed between t0 and t2 and in blood insulin between t1 and t4 after MCT ingestion alone. Thereafter, blood glucose remained stable, slightly above the baseline level, which implied a systemic homeostasis of gluconeogenesis [44]. Similar results were reported by other authors as well [13,38,39,41]. 

Similar to St-Pierre et al., the current study demonstrated that MCT ingestion with and without glucose resulted in higher plasma C8 and C10 levels over the 5 h test day, suggesting that spot checks of plasma C8 and C10 could be useful for monitoring compliance when one or more of these fatty acids are consumed in a longer-term study [39]. It is known that, upon supplementation, MCT is hydrolyzed in the gut to glycerol and medium-chain fatty acids (e.g., C8, C10) [37]. When released from the intestinal mucosa, MCFAs weakly bind to albumin [8]. The proportion between albumin-bound and free MCFAs increases with increasing chain length [8]. The uptake of MCFAs by the liver and other tissues is independent of fatty acid binding proteins, fatty acid transport proteins, and transmembrane translocation [45]. MCFAs are rapidly absorbed into the bloodstream and increase plasma concentrations of C8 and C10, which are transported to target organs and rapidly oxidized in the liver to CO_2_ and H_2_O in the citrate cycle to generate energy in the form of ATP via the mitochondrial OXPHOS system or biosynthesis of ketones [46,47,48]. The appearance of octanoic and decanoic acids in the blood hours after ingestion suggests that there may be alternate routes for MCFAs beyond beta oxidation and ketogenesis [44,46]. One recent study has reported that hepatocytes are able to incorporate MCFAs into triglycerides mainly destined for beta-oxidation and, to a lesser degree for incorporation into low-density lipoprotein particles [49]. Another study identified 73 paths involving 84 unique metabolites interconnected by 91 unique reactions during transformation of C8:0 to βHB, and an equally complex metabolism for C10:0 [50].

We did not measure total triglycerides in the study; therefore, further trials should explore the short- to long-term effects of MCT intake on postprandial lipid parameters such as fasting and postprandial TG and chylomicron triglycerides to support this evidence. 

Since this study measured levels of octanoic acid, decanoic acid, and ketones for up to five hours, we could not determine when the levels began to decrease and returned to baseline. Results of this study and previous studies suggest that an 8-h follow-up test period or longer and studying two or more daily doses could provide more complete information on the kinetics of BHB, C8, and C10 following MCT ingestion [41]. Furthermore, if MCT oils with less ketogenic MCFAs C10 and C12 are studied, it would make sense to monitor compliance by measuring the relevant fatty acid levels rather than BHB [39]. Sustained elevation of MCFAs in the blood following ingestion could be advantageous for persons consuming a ketogenic diet since the MCFAs would be constantly available to cells that can use MCFAs as fuel, such as astrocytes, skeletal, and cardiac muscle. Including more MCFAs in the diet in place of LCFAs could help improve glucose tolerance and reduce the tendency of high-fat diets with LCFAs to induce insulin resistance [51]. 

MCTs have repeatedly been shown to improve memory and benefit cognitive performance in healthy and diseased populations [52,53,54]. The brain can utilize BHB as an alternative energy substrate to glucose, and medium-chain fatty acids can directly cross the blood-brain barrier and be oxidized in astrocytes [55]. A recent study reported that when BHB and MCFAs are both present, astrocytes prefer MCFAs and neurons prefer BHB as fuel; furthermore, it has been proposed that astrocytes may shuttle BHB from C8:0 to neurons [50,56]. Furthermore, a study by Myette-Côté et al. reported an improved insulin sensitivity index with lower glucose levels and glucose area under the curve (AUC), suggesting improved glucose utilization following consumption of a single dose of BHB ketone ester compared to a placebo during an oral glucose tolerance test using 75 g of glucose in 20 healthy adults ages 18 to 35 [57]. In the present study, results showed a significant improvement in cognitive tests compared to the baseline scores after consuming MCT plus glucose. One scientifically reasonable mechanism that could explain this pro-cognitive effect is an augmentation of the brain’s energy supply in the form of glucose, BHB, and medium-chain fatty acids, which are all rapidly available to the brain as fuel [52]. We did not observe any correlations between BHB concentrations and either cognitive test, and, as an alternative explanation, the cognitive improvement following consumption of MCT and glucose might be due to practice and retest effects [58,59]. When effects of MCT supplementation on cognitive performance are a target of interest, neuropsychological tests without significant practice effects, e.g., the trail-making test, might be considered [60].

## 5. Conclusions

This study compared the ketogenic effect of two MCT oil interventions, with and without glucose, with the aim of assessing differential effects. In summary, we show in young, healthy adults that MCT oil supplementation alone and in combination with a small amount of glucose increased average BHB levels up to physiologically significant concentrations (>0.5 mM), along with a moderate increase in blood glucose and a large increase in insulin levels when glucose was added. MCT intervention alone resulted in a more rapid BHB response than MCT plus glucose but with similar final BHB concentrations. Plasma C8 and C10 levels were detected after both MCT interventions, suggesting that if MCT oils with less ketogenic MCFAs, e.g., only C10 and or C12, are studied, compliance could be monitored by measuring the relevant fatty acid levels during planned study visits in addition to BHB. Kinetic studies of longer duration are needed to determine how long BHB and individual MCFAs are detectable in the blood, which could help with dosing regimens when used for clinical indications and also for future study design. For MCT plus glucose consumption, we found fewer side effects and positive effects on cognitive ability, although statistical trends were also observed for MCT alone. Longer-term studies is needed to understand how MCT supplementation of meals with and without carbohydrate affects metabolic health, cognition, and other clinical outcomes in the long term for children and adults of all ages.

## Figures and Tables

**Figure 1 nutrients-15-01148-f001:**
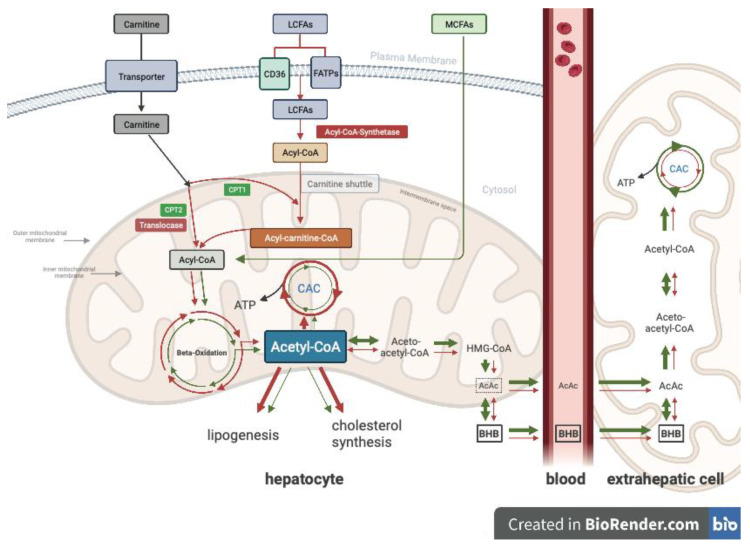
Simplified illustration of medium-chain fatty acid metabolism (represented by the green line).

**Figure 2 nutrients-15-01148-f002:**
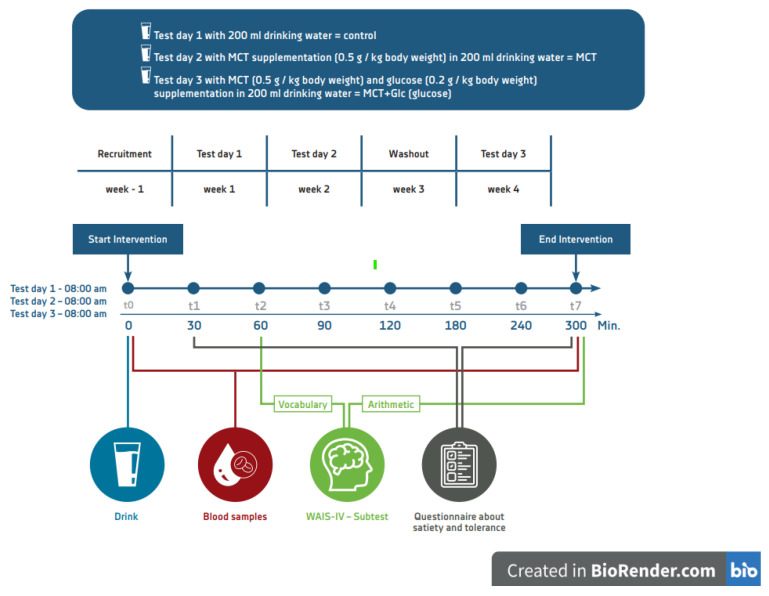
Overview of the study design.

**Figure 3 nutrients-15-01148-f003:**
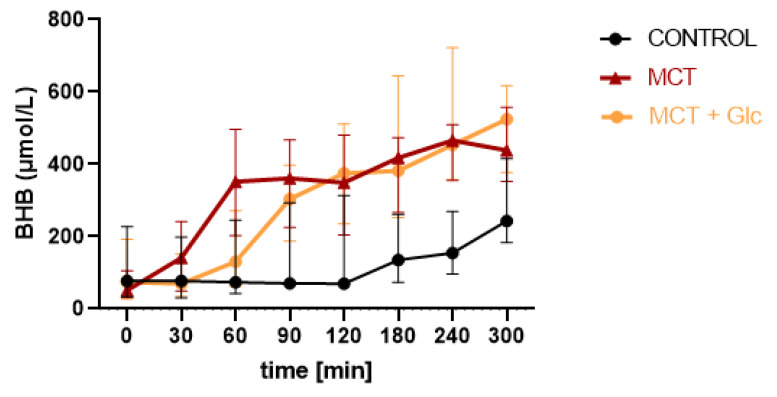
Median (IQR) beta-hydroxybutyrate levels throughout the 5 h test day.

**Figure 4 nutrients-15-01148-f004:**
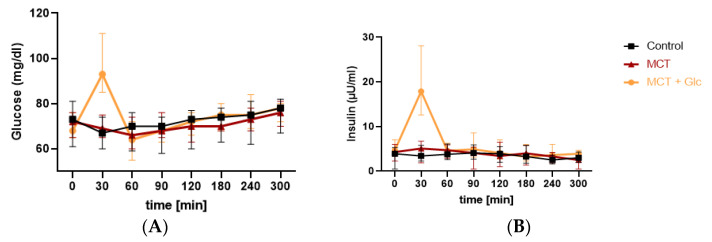
Median (IQR) glucose and insulin levels: (**A**) glucose levels (mg/dL) throughout the 5 h test day; (**B**) insulin levels (µU/mL) throughout the 5 h test day.

**Figure 5 nutrients-15-01148-f005:**
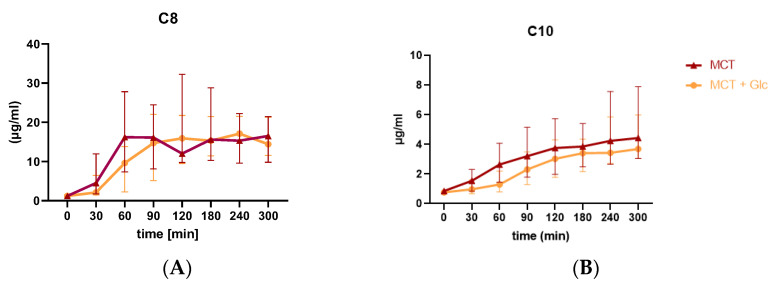
Median (IQR) plasma MCFAs: (**A**) plasma octanoate throughout the 5 h test day; (**B**) plasma decanoate throughout the 5 h test.

**Figure 6 nutrients-15-01148-f006:**
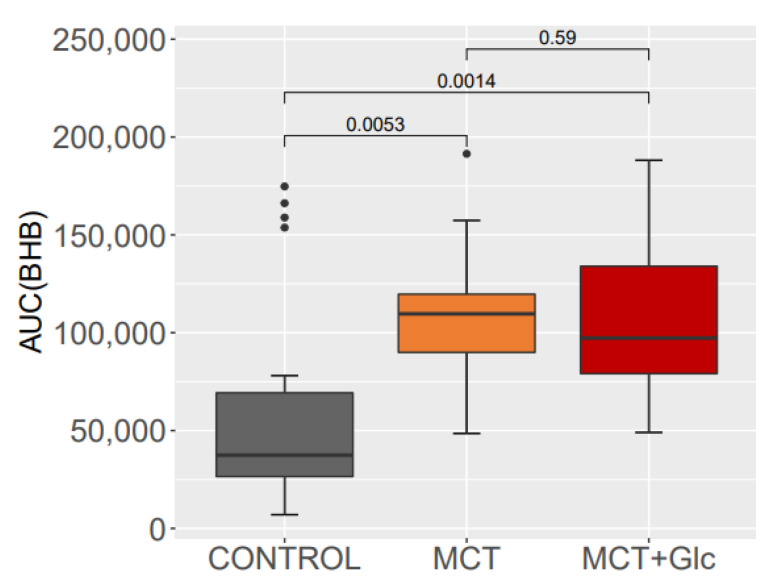
Area-under-the-curve (µg/mL * h/L) for plasma BHB during the five test days (control, MCT, and MCT plus glucose intervention).

**Figure 7 nutrients-15-01148-f007:**
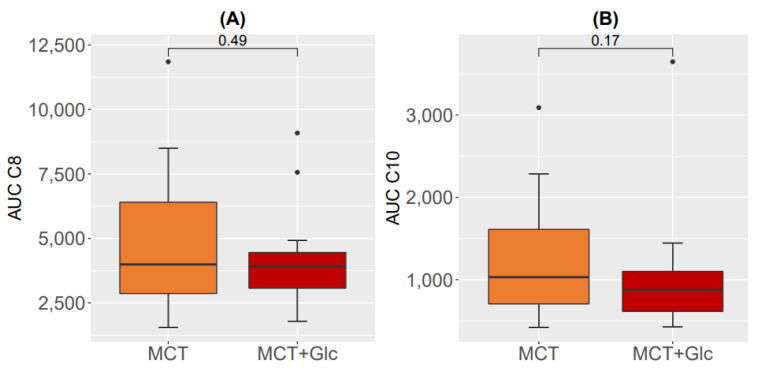
Differences in AUCs (µg/mL * h/L) for plasma MCFA: (**A**) plasma octanoate after MCT oil alone and after MCT oil plus glucose; (**B**) plasma decanoate after MCT oil alone and MCT oil plus glucose.

**Figure 8 nutrients-15-01148-f008:**
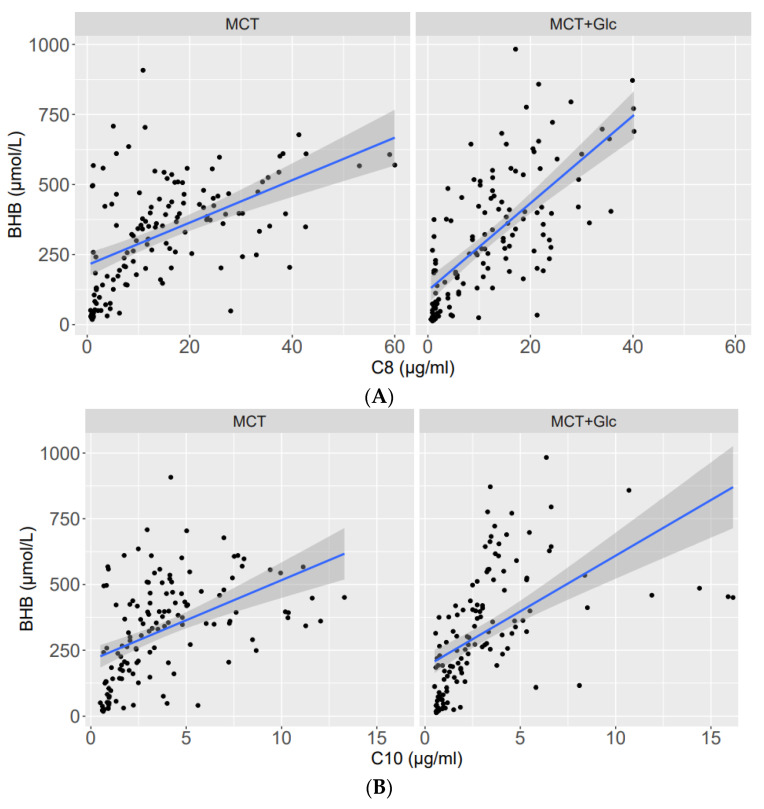
(**A**): C8—related plasma BHB levels with MCT (**left**) and MCT plus glucose (**right**). The indicated line represents the best fit of a linear regression with plasma C8 (µg/mL). (**B**)**:** C10—related plasma BHB levels with MCT (**left**) and MCT plus glucose (**right**). The indicated line represents the best fit of a linear regression with plasma C10 (µg/mL).

**Figure 9 nutrients-15-01148-f009:**
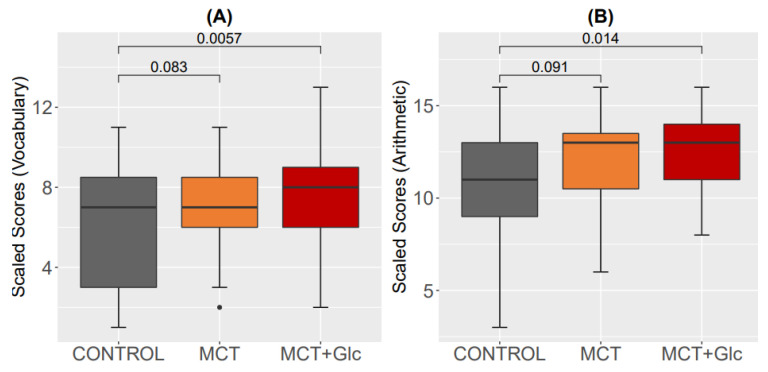
Results of subtests throughout the 5 h test day: (**A**) Vocabulary subtest; (**B**) Arithmetic subtest.

**Table 1 nutrients-15-01148-t001:** Baseline demographic and biochemical variables of the participants.

Characteristics		Values
n	(female/male)	19 (12/7)
age	years	25 (20–27) *
weight	kg	64 (61–71) *
height	cm	173.2 (169–177) *
BMI	kg/m^2^	22 (20–23) *
BHB	µmol/L	77 (34–227) *
fasting glucose	mg/dL	85 (79–89) *
fasting insulin	µU/mL	3.9 (0.5–5.3) *

* Values are presented as the median and interquartile range (IQR).

**Table 2 nutrients-15-01148-t002:** Side effects between t1 and t7 (*n* = 19) *.

Side Effects (SE)	MCT Oil	MCT Oil Plus Glucose
Total SE (n)		
t1	20	10
t7	0	0
Nausea		
t1	5	0
t7	0	0
Diarrhea		
t1	2	2
t7	0	0
Stomach discomfort		
t1	4	4
t7	0	0
Abdominal pain		
t1	4	4
t7	0	0
Dizziness		
t1	2	0
t7	0	0
Sweating		
t1	1	0
t7	0	0
Brain fog		
t1	1	0
t7	0	0
Vomiting		
t1	1	0
t7	0	0

* Participants (*n* = 5) reported several side effects; *n*: number of subjects. Time points (t1) and (t7).

## Data Availability

Not applicable.

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
