# Peer review of "Beta-Hydroxybutyrate (BHB), Glucose, Insulin, Octanoate (C8), and Decanoate (C10) Responses to a Medium-Chain Triglyceride (MCT) Oil with and without Glucose: A Single-Center Study in Healthy Adults"

_nutrients, 2023, doi:10.3390/nu15051148_

Round 1

Reviewer 1 Report

This study compared the ketogenic effect of two MCT oil interventions, with and without glucose, which is interesting and meaningful research. It could be proceed in our journal. Some of my questions are as follows after evaluating this manuscript.

1.      In your “Cognitive test”, this is an interesting part, why did you choose "vocabulary" and "arithmetic" and not others?

2.      The results in Figure 9 are interesting. For example, “For the arithmetic subtest, there was no significant difference between the control and the MCT oil alone intervention. However, a significant difference was found between the control and the MCT oil plus glucose intervention (p = 0.027).” I'm curious as to why the discrepancy occurs. Based on your description, I think it should be easy for any young person between the ages of 18-30 to come to the right conclusion. Have you considered that it is because they have different levels of education?

3.      In the discussion section, the links before and after should be more natural, and some of the reasoning and assertions should be more thoughtful.

Author Response

Dear Reviewer,

thank you for your report and thoughtful comments on our manuscript.

Please see below our point-by-point response and the revised article attached.

Thank you for your second review in advance!

With best wishes,

Christina 

____________________________________________________________

Point 1: In your “Cognitive test”, this is an interesting part, why did you choose "vocabulary" and "arithmetic" and not others?

Response 1: First, thank you very much for your thoughtful comments. 

Vocabulary and arithmetic subtests were chosen, because they are close to everyday cognitive performance, easy to administer, both loading high on their underlying factors “verbal comprehension” (vocabulary) and “working memory” (arithmetic), with arithmetic being largely independent from verbal comprehension according to Ward et al. [26].

We have included these information as well to the article, highlighted in red; please see line 172.

Point 2 A: The results in Figure 9 are interesting. For example, “For the arithmetic subtest, there was no significant difference between the control and the MCT oil alone intervention. However, a significant difference was found between the control and the MCT oil plus glucose intervention (p = 0.027).” I'm curious as to why the discrepancy occurs.

Response 2 A: You are right, this is a transmission error. Therefore, in response to your comment, we have updated the p-value; please see line 330.

Point 2 B: Based on your description, I think it should be easy for any young person between the ages of 18-30 to come to the right conclusion. Have you considered that it is because they have different levels of education?

Response 2 B: As the design was repeated measurement the same subjects were tested under the different conditions, why an effect of educational level can be excluded.

Point 3: In the discussion section, the links before and after should be more natural, and some of the reasoning and assertions should be more thoughtful.

Response 3: According to your suggestions, we have updated the discussion and conclusion part including glucose and MCT alone intervention; please see line 441 – 452 and 466-467.

Reviewer 2 Report

In this manuscript, Heidt et al. reported a human volunteer study of multiple metabolites in response to medium-chain triglyceride oil intake complexed with glucose intake. The metabolites biochemistry tests include several key biomarkers in human hyperglycemia, hyperlipidemia and, most importantly, ketogenic responses. Human behavior responses on cognitive functions were also documented under different treatment. 

The target study group is 19 young adults with a relatively narrow age range and thus, provided a focused study of MCTs/glucose response in this age group. The fact that the conclusion from this research is derived from the young adults age group should be clearly pointed out in the discussion and conclusions. Also, this age factor should be considered when comparing with research results from other labs. 

Given the great social impact of MCTs and ketogenic diets, this research is significant and meets the scope of Nutrients. The methods and analyses in this manuscript are straightforward and well documented. These rigorous factors supported the quality of this work. Overall, this work has high quality and meet the proceed standard of Nutrients.

Author Response

Dear Reviewer,

thank you for your report and thoughtful comments on our manuscript.

Please see below our point-by-point response and the revised article attached.

Thank you for your second review in advance!

With best wishes,

Christina 

_________________________________________________________________________________

Point 1: The target study group is 19 young adults with a relatively narrow age range and thus, provided a focused study of MCTs/glucose response in this age group. The fact that the conclusion from this research is derived from the young adults age group should be clearly pointed out in the discussion and conclusions. Also, this age factor should be considered when comparing with research results from other labs. 

Response 1: First, thank you very much for your thoughtful comments. According to your suggestions, we have updated the discussion and conclusion part; please see line 358, 362-363, 366-367, 371-374, 382-383, 456 and 469-470.

Point 2: Given the great social impact of MCTs and ketogenic diets, this research is significant and meets the scope of Nutrients. The methods and analyses in this manuscript are straightforward and well documented. These rigorous factors supported the quality of this work. Overall, this work has high quality and meet the proceed standard of Nutrients.

Response 2: Thank you for your constructive feedback to our group!

Reviewer 3 Report

I congratulate the authors for this excellent work.

Here are my little suggestions to improve the work:

1-Line 38, please explain the acronym 'FATPs'

2-line 81, insert non-obese and non-overweight because you used 25 BMI as a cut-off, not 30.

3-line 190, please insert the nutritional analysis of pre-study meals and drinks in the supplementary materials.

Author Response

Dear Reviewer,

thank you for your report and thoughtful comments on our manuscript.

Please see below our point-by-point response and the revised article attached.

Thank you for your second review in advance!

With best wishes,

Christina 

___________________________________________________________________________

I congratulate the authors for this excellent work.

Response: First, thank you very much for your thoughtful comments.

Here are my little suggestions to improve the work:

Point 1: Line 38, please explain the acronym 'FATPs'

Response 1: Thank you for this. According to your suggestion, we have included the abbreviation for FATPs (FATPs: fatty acid transport proteins); please see line 38.

Point 2: line 81, insert non-obese and non-overweight because you used 25 BMI as a cut-off, not 30.

Response 2: Thank you for this. According to your suggestion, we have inserted this information; please see line 78.

Point 3: line 190, please insert the nutritional analysis of pre-study meals and drinks in the supplementary materials.

Response 3: We have included the requested data; please see Supplementary Materials, Table S1 and in line 197 and 474.
